

# River discharge impacts coastal Southeastern Tropical Atlantic sea surface temperature and circulation: a model-based analysis

Léo C. Aroucha[1], Joke F. Lübbecke[1], Peter Brandt[1,2], Franziska U. Schwarzkopf[1], Arne Biastoch[1,2]

[1]GEOMAR Helmholtz Centre for Ocean Research Kiel, Kiel, Germany

5  [2]Faculty of Mathematics and Natural Sciences, Kiel University, Kiel, Germany

*Correspondence to*: Léo C. Aroucha (leo.aroucha@geomar.de)





**Abstract.** The Southeastern Tropical Atlantic (SETA) coastal region sustains highly productive fisheries and marine ecosystems, thus having immeasurable socio-economic importance for West African countries. It is characterized by high sea surface temperature (SST) variability and freshwater input from land mainly due to Congo River discharge. In this study, using high-resolution ocean model sensitivity experiments, we show that the presence of low salinity waters from the river discharge increases the mean state SST in the SETA coastal fringe by about 0.26ºC on average and by up to 0.9ºC from south of the Congo River to the Angola-Benguela front (ABF). North of the Congo River up to about 4ºS, this input significantly reduces the mean state SST by more than 1ºC. We demonstrate that the impact of river discharge on SST is associated with a halosteric effect, which modifies the sea surface height gradient and alters geostrophic currents, producing a southward (northward) coastal geostrophic flow, with an onshore (offshore) geostrophic component to the south (north) of the Congo River. Hence, advective warming (cooling) and downwelling (upwelling) are generated south (north) of the river mouth. Furthermore, the southward advection generated by the low salinity waters pushes the ABF further south. These results draw attention to the freshwater impact on SSTs and ocean surface dynamics, especially in the projected climate change scenario of continuously increasing land to ocean discharge.

## 1 Introduction

The Southeastern Tropical Atlantic (SETA, Fig. 1) is an eastern boundary upwelling region that features high biological productivity encompassing the tropical Angolan, and northern and southern Benguela upwelling systems (Jarre et al., 2015). This area sustains productive marine ecosystems and fisheries, with high socio-economic and food security importance for West African countries (FAO, 2020; Kirkman et al., 2016; Sowman and Cardoso, 2010). Within the SETA, the thermal Angola-Benguela front (ABF, Fig. 1a), which is characterized by the convergence of tropical warm waters from the north and cool subtropical waters from the south, is located at around 17ºS-18ºS, where the southward Angola Current (AC) encounters the northward Benguela Coastal Current (BCC) (Koseki et al., 2019). Due to the strong meridional sea surface temperature (SST) gradient in the ABF, its migration to the north and south is decisive for SST changes off Angola, and plays a major role in determining seasonal and interannual SST variability in the region (Lübbecke et al., 2010).

The source of riverine freshwater input in the SETA involves the outflow of three West African rivers (Fig. 1): the Congo River, with its mouth at around 6ºS; the Cuanza River at about 9ºS; and the Kunene River at 17ºS. The major input comes by far from the Congo River, which is the second largest river system in the world and has a mean flow rate of about 40.000 m$^3$s$^{-1}$ (Dai and Trenberth, 2002; Campbell, 2005). The Congo River discharge (CRD) peaks during November-January, has a secondary maximum in April-May, and minima during March and August (Fig. S1). The discharge seasonality is governed by the precipitation over the Congo River basin which is mainly related to the movement of the Intertropical Convergence Zone (ITCZ) (Sorí et al., 2017; Munzimi et al., 2019). The CRD is the greatest contributor to the sea surface salinity (SSS) mean state and variability in the eastern Tropical Atlantic (Martins and Stammer, 2022; Chao et al., 2015; Denamiel et al., 2013;



Hopkins et al., 2013; Materia et al., 2012). Recently, both the CRD and the SSS in the area were linked to the occurrence of
the Indian Ocean Dipole (IOD) with positive IOD events leading to increased moisture convergence over the Congo basin and
subsequently enhanced CRD (Jarugula and McPhaden, 2023). The Congo River plume and the low SSS signal spread usually
westward, driven mainly by zonal advection (Houndegnonto et al., 2021; Martins and Stammer, 2022). From February to
April, however, the plume reaches its seasonal southernmost extension up to 12ºS, controlled by the meridional advection of

the AC (Kopte et al., 2017; Awo et al., 2022; Martins and Stammer, 2022). In some specific years, the low SSS signal can
reach as far south as 18ºS (Aroucha et al., 2024; Gammelsrød et al., 1998).

River related low SSS has been associated with increased SST in the Eastern Tropical Atlantic. Materia et al. (2012) used
observations to correlate freshwater discharge and SST increase in the Gulf of Guinea via mixed layer shoaling, while Martins

and Stammer (2022) showed that the low SSS waters associated with CRD strongly increase stratification in the region.
Moreover, it has been recently shown that freshwater input also contributes to boosting strong anomalously warm coastal
events off the Angola coast, the so-called Benguela Niños (Shannon et al., 1986; Florenchie et al., 2004), via reduced turbulent
heat loss due to increased stratification (Lübbecke et al., 2019; Aroucha et al., 2024). One of the main conditions for the low
SSS influence on these events is that the freshwater plume from the CRD is advected to the Angolan-Namibian coasts, much

further south than climatologically expected (Awo et al., 2022). These studies are based on the mechanism of the low salinity
shoaling the mixed layer, which generates barrier layers between the density-stratified mixed layer and the temperature-
stratified isothermal layer. The barrier layer presence weakens the vertical temperature gradient between the mixed layer and
the waters below it, reducing the impact of vertical mixing and turbulent heat loss, which then contributes to increasing SSTs.

On the other hand, the modelling study by White and Toumi (2014) did not find a substantial influence of the barrier layer
generation on increasing the SST mean state off the Congo River mouth. By using different simulations with and without the
river presence, the authors showed that although the CRD generates barrier layers up to 6m thick, the shoaling of the mixed
layer is such that the penetration of the solar shortwave radiation is not trapped within this layer. Hence, the cooling effect
from the reduced shortwave absorption exceeds the warming impact of the reduced vertical mixing (White and Toumi, 2014).

However, their simulations showed a significant coastal warming impact from the CRD south of the river mouth (Fig. 5 in
White and Toumi, 2014). At the same time, model simulations have shown that freshwater input and variability can indeed
increase the mean state SST in other tropical regions. Zhang and Busalacchi (2009) pointed to the increase of positive SST
anomalies due to amplified freshwater fluxes in the Tropical Pacific, while Topé et al. (2023) showed a similar effect in the
Gulf of Guinea due to the presence of the Niger River. It has been demonstrated that the Niger River can also limit coastal

upwelling in the Gulf of Guinea through the generation of an onshore geostrophic flow (Alory et al., 2021). Additionally,
recent climate projections have identified changes in the onshore geostrophic flow as a key factor controlling long-term trends
in eastern boundary upwelling systems (EBUS) (Jing et al., 2023), highlighting the potential influence of the CRD on the
future dynamics of the SETA upwelling system.



Overall, it is clear that river plumes affect not only the SST mean state but also its variability in tropical regions. For the SETA, specifically, they can amplify extreme warm events, impacting fisheries, marine ecosystems, and the socio-economics of African countries. Hence, it is crucial to better recognize the mechanisms by which freshwater input influences SST changes. This comprehension becomes even more important in the future warming scenario, which has been shown to amplify river runoff around the world (Müller et al., 2024; Aloysius and Saiers, 2017). In the present study, we aim to understand the impact

of the freshwater input especially on the mean state SST, focusing on the SETA region. For this, we use three sensitive experiments from a nested ocean general circulation model: a control experiment with an interannual varying freshwater discharge from land to ocean (CTRL); a sensitivity experiment with a climatological freshwater runoff (CLIMA); and one with no runoff at all (NORIV). More details are given in Sect. 2.

This paper is organized as follows: in Sect. 2 we present the model configuration, the experiments, the datasets, and the methods for upwelling indices, advection term, and barrier layer calculation; in Sect. 3 we compare the model output against observational datasets, describe the differences between the experiments and the mechanisms responsible for these differences; in Sect. 4 we discuss our main findings and present concluding remarks.

## 2 Model, datasets and methods

### 2.1 INALT20

In this study we used the INALT20 model configuration (Schwarzkopf et al., 2019), which is based on the NEMO v3.6 ocean general circulation model (Madec and the NEMO team, 2016). It consists of a high-resolution (1/20°) nest covering the South Atlantic and the West Indian oceans (70°W – 70°E, 63°S – 10°N, Fig. 2 in Schwarzkopf et al., 2019), embedded into the coarser resolution (1/4°) global host grid ORCA025 via a two-way nesting approach, allowing the host to provide boundary conditions

to the nest as well as receiving information from the nest. INALT20 has a vertical grid consisting of 46 z-levels, with 6 m resolution near the surface and 14 levels in the upper 200 m. Although the uppermost level represents a 6 m thick layer with horizontal velocities, temperature and salinity defined at its centre (~3 m depth), we consider those as surface values. More details on INALT20 can be found in Schwarzkopf et al. (2019). The model is forced at the surface with momentum, heat and freshwater fluxes from the atmospheric product JRA55-do (Tsujino et al., 2018). It has been recently shown that the use of

JRA55-do as the forcing product performs better in the Benguela upwelling region in comparison to other datasets (Small et al., 2024).

### 2.1.1 Model experiments

The study is based on three simulations that only differ in their prescribed river runoff. In the reference (control) simulation (CTRL) an interannually varying daily runoff forcing from JRA55-do (Tsujino et al., 2018) is applied. This simulation has



been previously used and described also in Schmidt et al. (2021), Biastoch et al. (2021) (therein referred to as INALT20-JRA-long), and Rühs et al. (2022) (SIM$_{JRA}$). The sensitivity experiment CLIMA is forced with a monthly climatology of the years 2000 to 2019 built from the JRA55-do runoff. In the second sensitivity experiment (NORIV) no runoff forcing was applied. For CLIMA and CTRL, vertical mixing is enhanced where runoff enters the ocean. The hindcast simulation from 1958 to 2019 (CTRL) is preceded by a 30-year long spin-up integration initialized with temperature and salinity data from the World Ocean Atlas (WOA) (Huang et al., 2021; Levitus et al., 1998). CLIMA and NORIV branch off from CTRL in 2000, spanning from the period 2000-2018. We analyze 5 day-averaged output data, except for sea surface height (SSH) and freshwater runoff, for which daily outputs are used. Modelled SSH fields are detrended by removing the global average at each location and time slice.

## 2.2 Observational and reanalysis datasets

To assess the model performance in representing the mean state and variability of key variables, we compared CTRL outputs against a variety of datasets ranging from satellite and reanalysis products to in situ measurements. The datasets are described below. We used monthly averages of all these products for the time period from 2000-2018 to have a consistent comparison with the corresponding model output. Exceptions are mentioned below. Finally, all gridded datasets were interpolated onto the INALT20 1/20° spatial grid. Here, interannual variability is calculated as the monthly standard deviation of the variable anomalies.

A blend of satellite and in situ measurements of SST was obtained from the high-resolution NOAA Optimum Interpolation SST (OISST) (Huang et al., 2021; Reynolds et al., 2007). The data has ¼° spatial resolution and is available from 1981 to the present day at the National Center for Environmental Information.

For salinity, we took SSS v03.21 measurements from the European Space Agency Sea Surface Salinity Climate Change Initiative (ESACCI) (Boutin et al., 2021). It consists of composites of bias-correct SSS from the satellite missions Soil Moisture and Ocean Salinity (SMOS, 2010-2019), NASA Aquarius (2012-2015), and Soil Moisture Active Passive (SMAP, 2015-present), with the data available from 2010 to 2018.

SST and SSS as well as zonal and meridional ocean velocities were obtained also from the GLORYS12 reanalysis product (Lellouche et al., 2021). It is based on the Nucleus for European Modelling of the Ocean (NEMO) with atmospheric forcing by ERA-Interim and ERA5 with assimilated in situ profiles of temperature and salinity from the CORAv4.1 database. It has a 1/12° horizontal resolution and 50 vertical levels and covers the period from 1993 to the present. The product is distributed by the EU Copernicus Marine Service (CMEMS). Although GLORYS12 is forced by climatological river runoff, it has shown good performance in the SETA region in reproducing the temperature and salinity mean state and variability both at the surface and at depth (Aroucha et al., 2024).



To validate the freshwater input data from the model, we use Congo River discharge data at Kinshasha-Brazzaville Station,
140   Republic of Congo from the ORE-HYBAM observatory. Daily values are available from 1947 to 2023.

Finally, we used current velocity measurements from 45m to 500m depth from a moored ADCP off Angola at 13ºE, 10º50'S,
77 km away from the coast (Kopte et al., 2017). The data is available from 2014 to 2021. Alongshore and cross-shore velocities
were derived by rotating 34º anticlockwise from the north. To complement the mooring measurements at the surface, we
obtained monthly total current velocities from the GLOBCURRENT dataset (Rio et al., 2014), which consists of zonal and
meridional velocities at the surface and 15m depth from combined CMEMS satellite geostrophic currents and modelled Ekman
currents. This dataset is available at 0.25º resolution from 1993-2022, and is also distributed by the CMEMS.

**2.3 Mixed layer depth, isothermal layer depth, and barrier layer thickness definitions**

The isothermal layer depth (ILD) is here defined by a 0.2ºC threshold referenced to the temperature value at 3m depth (first
vertical level of the model), $T_{3m}$, while the mixed layer depth (MLD) is the depth at where the potential density ($\sigma_0$) referenced
to its value at 3 m depth ($\sigma_{3m}$), is increased by an amount equivalent to a 0.2ºC temperature change at the local salinity ($\Delta\sigma_0$),
as follows:

$$ILD = depth\ where\ [T = T_{3m} - 0.2ºC] \tag{1}$$

$$MLD = depth\ where\ [\sigma_{0 =} \sigma_{3m} + \Delta\sigma_0] \tag{2}$$


$$\Delta\sigma_0 = \sigma_0(T_{3m} - 0.2ºC, S_{3m}, P_0) - \sigma_0(T_{3m}, S_{3m}, P_0) \tag{3}$$

where $S_{3m}$ and $P_0$ are salinity and pressure at 3m depth and the ocean surface, respectively. The barrier layer thickness (BLT)
is the difference between the ILD and MLD: $BLT = ILD - MLD$. These fields were calculated based on 5-day-averaged
temperature and salinity vertical profiles, and these definitions, which ensure that salinity changes at the surface are particularly
considered for shoaling the mixed layer and generating barrier layers, have been extensively used during the last years (e.g.
Aroucha et al., 2024; Gévaudan et al., 2021; Saha et al., 2021). In addition, the squared Brunt-Väisälä frequency ($N^2$) fields
were obtained by averaging from surface to 50m depth the $N^2$ at each depth level calculated from the monthly temperature and
salinity vertical profiles.

**2.4 Horizontal advection**

To investigate whether SST differences between the simulations stem from changes in surface dynamics and consequently
surface advection, we calculated the surface horizontal advection as:

$$Adv = -\rho c_p h\ \boldsymbol{v}.\nabla T \tag{4}$$




where $\boldsymbol{v}$ and $\nabla$T are the 5-day-averaged surface horizontal currents and temperature gradient, respectively, $\rho$ is the water

density, here taken as 1025 kg/m³, $c_p$ is the specific heat capacity, equal to 4000 J/kg °C, and $h$ is the surface level representing

a 6 m thick layer. Advection was calculated individually for each experiment, where $\boldsymbol{v}$ and $\nabla$T were both taken from the same

experiment (e.g. $Adv_{CLIMA} = -\boldsymbol{v}_{CLIMA}.\nabla T_{CLIMA}$). Values within 20km off the coast were neglected.

**2.5 Upwelling indices**

Following Alory et al. (2021) and Marchesiello and Estrade (2010), we define dynamical upwelling indices to evaluate the

freshwater input impact on the competing effects of geostrophic flow and Ekman transport on coastal upwelling. The effect of

convergence/divergence of the geostrophic flow at the coast on vertical velocities is described by the Geostrophic Coastal

Upwelling Index (GCUI):

$$GCUI = \frac{-u_G \cdot MLD}{2L_u} \tag{5}$$

where $L_u$ represents the cross-shore width where upwelling occurs, here defined as 50km (see Fig. 5), $u_G$ is the cross-shore

surface geostrophic current (defined below) averaged within the $L_u$ and MLD is the mixed layer depth. Defining $L_u$ as 50km

fits to the width of the minimum cross-shore temperature gradient in this area, which is located at the shelf break (Körner et

al., 2023), and is close to the Rossby radius of deformation in the region (Chelton et al., 1998).

The Ekman Coastal Upwelling Index (ECUI) is a function of $L_u$, the alongshore wind stress, $\tau_a$, averaged within $L_u$, the water

density, $\rho$, and the Coriolis parameter, $f$, and is defined as follows:

$$ECUI = \frac{-\tau_a}{\rho f L_u} \tag{6}$$


The ECUI represents the vertical velocities due to the convergence/divergence of the Ekman transport at the coast. Positive

(negative) values of both GCUI and ECUI represent upward (downward) vertical velocities, indicating upwelling

(downwelling). The total Coastal Upwelling Index is the sum of both indices: $CUI = ECUI + GCUI$.

The surface cross-shore ($u_G$) and alongshore ($v_G$) geostrophic currents were calculated from the sea surface height (SSH) fields

based on the following equations:

$$v_G = \frac{g}{f} \frac{\partial SSH}{\partial x} \tag{7}$$

$$u_G = -\frac{g}{f} \frac{\partial SSH}{\partial y} \tag{8}$$





where $g$ is the gravitational acceleration, here taken as 9.8 m/s$^2$, and $f$ is the Coriolis parameter. Both the geostrophic currents and the upwelling indices were calculated based on 5-day data averages and individually for each model experiment.

# 3 Results

## 3.1 Model validation

Figure 1a shows the CTRL mean SST for 2000-2018 in the SETA compared to the mean SST for the same period for both OISST and GLORYS12. Overall, the SST spatial mean state from CTRL compares well to the mean SST of both products (Fig. 1a), although CTRL overestimates the SST in almost the whole SETA (Fig. 1b and 1c). The highest and most considerable biases are within the ABF and the coastal upwelling regions (i.e. from 15ºS to 30ºS). In CTRL, the ABF is located too far south, creating a warm bias at 17ºS – 18ºS of around 2.5ºC and 2.0ºC compared to OISST (Fig. 1b) and GLORYS12 (Fig. 1c), respectively. In the Benguela upwelling region (~19ºS – 30ºS), the coastal warm biases to both products are of similar spatial pattern and magnitude.

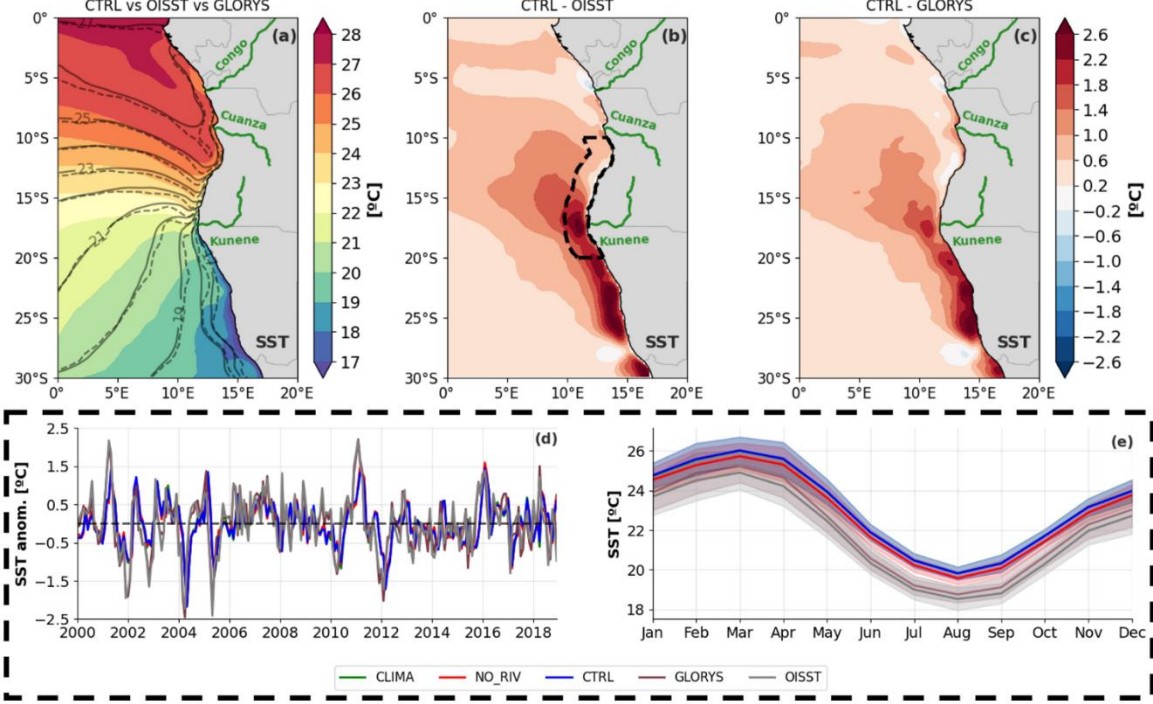

**Figure 1.** Comparison of simulated SST against observations and reanalysis. **(a)** Mean SST from CTRL (shading), OISST (solid contours), and GLORYS (dashed contours) averaged from 2000-2018. Contour interval is 1 ºC. **(b)** Difference between CTRL and OISST mean SST. Black dashed contour indicates coastal Angola-Benguela area (CABA, 10ºS–20ºS, 200km away from coast). **(c)** Difference between CTRL and GLORYS mean SST. **(d)** SST anomalies averaged for CABA shown for the three model experiments (CLIMA in green, NORIV in red, CTRL in blue) and datasets (GLORYS in brown and OISST in grey). **(e)** Same as (d) but for the monthly climatologies. Shadings indicate monthly standard deviations.





In the coastal Angola-Benguela area (CABA;10ºS–20ºS, 200km away from coast), CTRL depicts well the SST anomalies as presented by the satellite and the reanalysis product (Fig. 1d), although they are underestimated in specific years – especially during Benguela Niño events (e.g. 2001 and 2011). In fact, the SST variability from CTRL is lower within the whole SETA region (Fig. S2a-c) relative to both products, mainly in the CABA and ABF regions. The highest SST variability in the model

simulation is restricted to a thin coastal band, while the OISST and GLORYS12 highest variabilities are observed from 8ºE to the coast (Fig. S2a-c). Regarding the SST seasonality, all model experiments nicely agree with both OISST and GLORYS12 products (Fig. 1e). As previously mentioned, a warm bias is indeed present in the CABA, being stronger when compared to the satellite product than the reanalysis. It is noteworthy that for both CABA SST interannual variability and seasonality, there is no substantial difference between CTRL and CLIMA (blue lines mostly overlap green lines in Fig. 1d-e). The reasons for

this will be further discussed. However, in the seasonal climatologies, a constant offset of ~0.3ºC is observed between the simulations with (CTRL and CLIMA) and without freshwater input (NORIV) at the coastal Angola-Benguela region (Fig. 1e), already indicating that this input could create an SST mean state difference.

For the model validation with respect to SSS (Fig. 2) we focus on two different regions: the previously cited CABA region

(where the highest SST variability is present in the SETA, i.e. the region where Benguela Niños occur); and the Congo River mouth area (CRMA, solid black contour in Fig. 2b, 2ºS–10ºS, 200km away from coast), which represents the area of major freshwater input to the SETA. Overall, the SSS mean state from CTRL compares well against both ESACCI and GLORYS12 products with the lowest salinity values present within the CRMA, as a response to the CRD. However, there are SSS biases when comparing CTRL to both products. North of 10ºS and compared to ESACCI, CTRL slightly overestimates the offshore

SSS up to ~5ºS, while close to the coast it presents salinity values more than 1 psu lower than observed with satellites (Fig. 2b). North of 5ºS CTRL presents a fresh bias in relation to ESACCI, both at the coast and offshore (Fig. 2b). The fresh bias at the coast of CTRL compared to ESACCI occurs especially from October to May (Fig. 2g). On the other hand, when comparing the simulated SSS north of 10ºS to the GLORYS12 product, the model simulation shows mean coastal SSS values exceeding those of the reanalysis product by more than 1.5 psu, while there is a fresh bias west of 12ºE (Fig. 1c). At the coast, CTRL

presents higher SSS values than GLORYS12 mainly from June to September (Fig. 2g). Besides the observed differences, the SSS seasonal cycles at the CRMA are similar within the three products with the lowest (highest) salinity values coinciding with the season of highest (lowest) CRD – i.e. austral summer (winter) (Fig. 2g).

Regarding the CABA region, both ESACCI and GLORYS12 present similar SSS mean values and spatial patterns. When

comparing these products to CTRL, we see a salty bias ranging from 0.1 to 0.7 psu from 10ºS extending further south (Fig. 2b, c). It is an indication that the CRD freshwater influence on the SSS in CTRL is restricted to further north than what is observed in satellite and reanalysis products. The difference is even higher from September to November (Fig. 2e). As a consequence, the CTRL output presents a higher (lower) SSS variability north (south) of 10ºS in comparison to both products (Fig. S2d-f). The CTRL SSS variability is especially weaker in the CABA (Fig. 2d and Fig. S2d-f), which could partly explain




the same area of weaker CTRL SST variability, since SSS has been shown to influence extreme warm events in the CABA (Lübbecke et al., 2019; Aroucha et al., 2024). Finally, it is important to highlight that CTRL and CLIMA, similarly to what was observed for SST, show very similar SSS anomalies and seasonal climatologies, especially in the CABA (Fig.2d-g, blue and green lines overlapping).

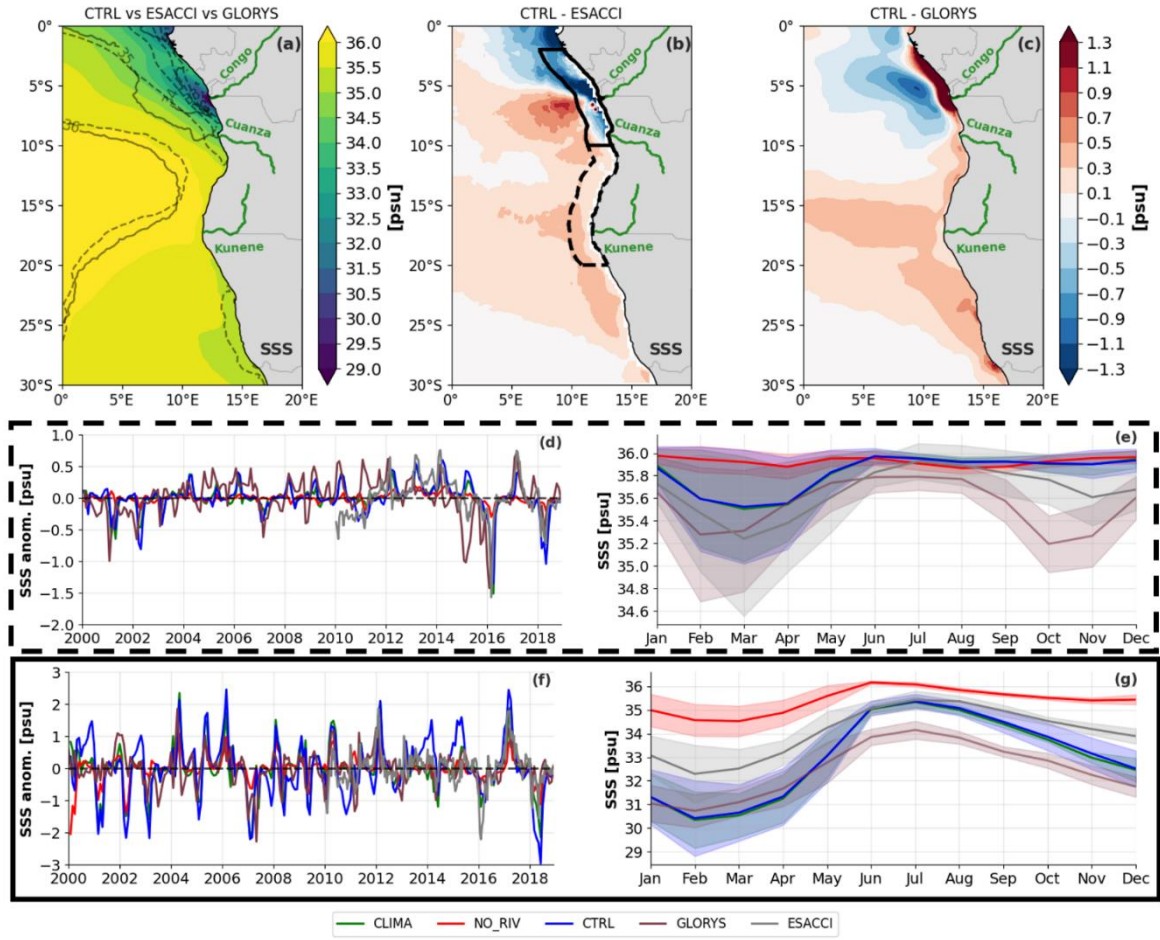


**Figure 2.** Comparison of simulated SSS against observations and reanalysis. **(a)** Mean SSS from CTRL (shading), ESACCI (solid contours), and GLORYS (dashed contours; contour interval is 1 psu). Both CTRL and GLORYS averaged from 2000-2018, while ESACCI averaged from 2010-2018. **(b)** Difference between CTRL and ESACCI mean SSS from 2010-2018. Black dashed contour indicates coastal Angola-Benguela area (CABA, 10ºS – 20ºS, 200km away from coast). Black solid contour shows Congo River Mouth area (CRMA, 2ºS-10ºS, 200km away from coast). **(c)** Difference CTRL and GLORYS mean SSS from 2000-2018. **(d)** SSS anomalies averaged for CABA for the three model experiments (CLIMA in green, NORIV in red, CTRL in blue) and datasets (GLORYS in brown and ESACCI in grey). **(e)** Same as (d) but for the seasonal climatologies. Shadings indicate monthly standard deviations. **(f)** and **(g)** same as (d) and (e) but for CRMA..

In terms of freshwater input, it is central to bear in mind that the model freshwater input considers not just the freshwater from the Congo River, but all freshwater input from land to the ocean, given by JRA55-do. To check how much of this input can be

attributed to the CRD, we averaged the CTRL freshwater input for a coastal box at the Congo mouth and compared it to the



discharge at the Kinshasha-Brazzaville station, which represents 98% of the total CRD (Alsdorf et al., 2016) (Fig. S1). In terms of magnitude, the seasonality of both freshwater inputs is similar (Fig. S1e). The seasonal climatology of the CTRL input compares well with the in situ station especially from December to April, while from June to October it shows lower freshwater input into the ocean (Fig. S1e). Although the seasonality is well represented by CTRL, the interannual variability of the
freshwater input is not as highly correlated to the Kinshasha-Brazzaville station (i.e. r = 0.46; Fig. S1d). Similar discrepancies were observed in the study by Chandanpurkar et al. (2022).

Furthermore, the alongshore and cross-shore current velocities from CTRL were checked against the 11°S mooring measurements taken in the CABA (Fig. S3). The mooring velocities were complemented at the surface with data from
GLOBCURRENT (see Sect. 2.2). In general, the AC is underestimated by CTRL throughout the year, especially from January to May (AC core located around 30-70 m depth) (Fig. S3b, d). Further, the model presents a much stronger northward surface coastal jet than what is observed at the mooring position. Since advection plays a major role in the SSS distribution in the region, it is likely that a stronger (weaker) than observed northward (southward) current traps the low SSS further north in CTRL. This would explain the increased (reduced) SSS variability north (south) of 10°S previously identified (Fig. S2d-f). It
would also mean that the freshwater transport to south of 12°S is nearly absent in the model experiments. Still, at the surface, CTRL seems to well represent the seasonality of the GLOBCURRENT velocities at the mooring position, with a stronger southward (northward) current from January to February (March to August) and September to October (November to December) (Fig. S3b,d).

Overall, we believe that CTRL resembles the major aspects of the mean state and seasonality of the variables discussed here. Moreover, the initial analysis of NORIV already points out the impact of freshwater input on both SST and SSS. Still, it seems that no significant differences result from interannual vs. climatological land to ocean discharges, indicating an insignificant impact of the interannual freshwater input variability on the variables analyzed here. In the following, we will analyze the differences between the three experiments.

**3.2 Impacts of freshwater input**

**3.2.1 Differences between the experiments**

To understand whether the freshwater input into the ocean impacts the SST near the West African coast, we first look at the mean SST differences between the three experiments (Fig. S4). The freshwater presence significantly affects the SST mean state in the coastal SETA region, regardless of using an interannual (CTRL) or climatological (CLIMA) runoff (Figs. S4a, b);
similar SST differences between these two experiments with discharge and the one without (NORIV) are observed not only in terms of the spatial pattern but also in the magnitude of those differences. At the same time, no significant mean SST difference was observed between CTRL and CLIMA experiments (Fig. S4c), as also seen in Fig. 1d and Fig. 1e. Additionally, CTRL and



CLIMA did not significantly differ in the mean SSS fields (Fig. S4f). Regarding SST variability, our experiments show no significant influence of freshwater input on monthly standard deviations of SST anomalies (Fig. S5). We thus conclude that
the presence of a land to ocean discharge alone generates the observed SST differences, with no further impact of an interannual varying discharge. Hence, in the following, we will focus on the effects of the climatological freshwater runoff presence on the SST mean state in the SETA.

Figure 3 depicts the mean state difference between CLIMA and NORIV for SST, SSS, SSH, ocean currents, and horizontal
advection (Adv). The freshwater input significantly warms the West African coastal fringe, south of the Congo River mouth. By including the discharge, the mean SST is increased by up to 0.9 ℃ near the coast at about 13ºS, and 0.26 ℃ on average in the CABA (Fig.3a-d). With freshwater input, strong warming is also observed at the Kunene River mouth (Fig. 3c). This pattern of positive SST differences is observed along the whole coast, from the Congo River mouth at ~6ºS extending to ~28ºS. Concomitantly, freshwater discharge generates a significant cooling north of the Congo River (i.e. decrease in SST of up to -
1.9℃ at 6ºS), with a maximum of negative SST difference from ~6ºS to ~4ºS (Fig. 3a).

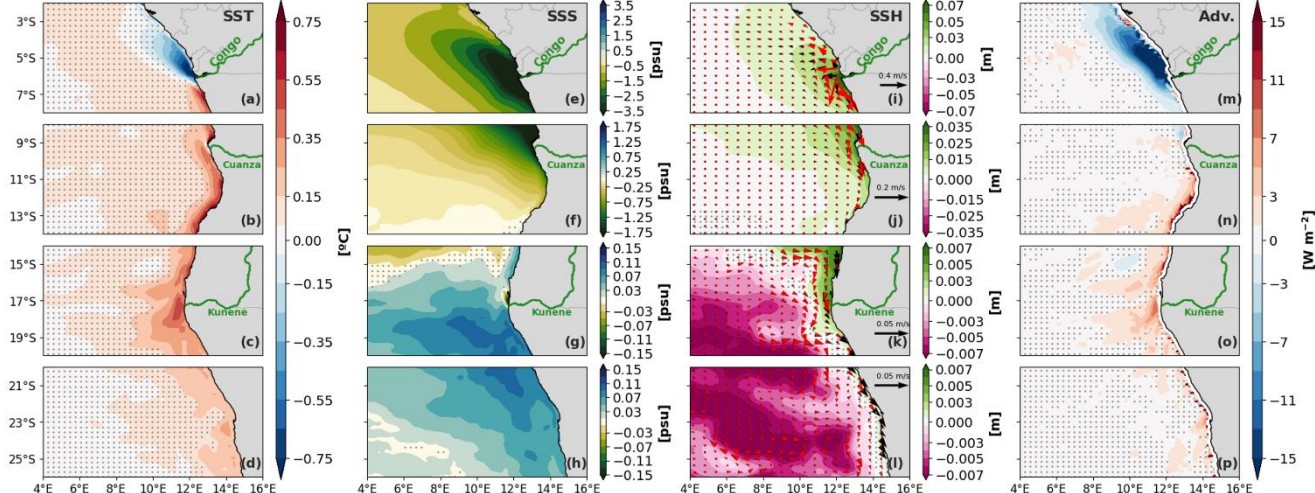

**Figure 3.** Differences between CLIMA and NORIV mean states (CLIMA-NORIV) for SST **(a-d)**; SSS**(e-h)**; SSH**(i-l)**; Horizontal Advection **(m-p).** Stippled grey areas indicate where the difference is not significant in a 95% confidence level. Black (red) arrows from i-l depicts total
(geostrophic) currents difference between experiments. Note that colorscales change for individual subplots for SSS and SSH.

In order to understand this difference in SST, we investigate the dynamic effects of the river presence. A land to ocean discharge significantly reduces the SSS in the whole SETA up to ~15ºS (Fig. 3e-g). The magnitude of freshening is stronger at the Congo's mouth where differences reach -6 psu and decreases towards the south (note that colorscale limits change from
Fig. 3e to Fig. 3g). Although different rivers flow into the Atlantic at the West African coast, the discharge rate from the Congo River is by far the largest and completely dominates the SSS differences observed between the model simulations. Still, a




lower magnitude SSS reduction is also observed at the Kunene River mouth (Fig.3g). The mean low SSS plume from the
Congo River spreads towards the west-northwest, consistent with the main direction of the CRD dispersion shown in previous
studies (Houndegnonto et al., 2021; Awo et al., 2022; Martins and Stammer, 2022). From 17ºS to the south, however, an
overall increase in SSS is noted when including the discharge in the experiments. Even though the magnitude of the SSS
increase is only ~5% of the SSS decrease further north, it is still significant (Fig. 3g-h). An SSS decrease due to the freshwater
presence is accompanied by a similar pattern of increasing SSH (Fig.3i-l). This inverse relation is a consequence of the
halosteric effect in the water volume, i.e. an expansion (contraction) of a water volume via the density reduction (increase)
owing to the lower (higher) salinities. The SSH increase patterns in the SETA coincide not only with the strong shift in SSS
due to the CRD, but also with the regions at the coast, where the model's freshwater input into the ocean takes place (Fig. S1).
Differences in SSH reach up to 7cm at the Congo's mouth (i.e. ~6ºS, Fig. 3i), and are ~10 times smaller at Kunene River
mouth, although still significant (Fig. 3k).

As a consequence of the changes in SSH, the freshwater input changes the dynamics of the coastal SETA. The red (black)
arrows depicted in Fig.3i-l represent the difference between CLIMA and NORIV in the geostrophic (total) currents. By
including the land to ocean discharge, a strong coastal southward jet is generated at the West African coast from south of 6ºS.
This current is mainly geostrophic and follows the SSH gradients to ~17ºS. South of this latitude the ageostrophic component
is stronger, but geostrophy still seems to play a role in the southward current (Fig. 3l). In contrast, north of the Congo's mouth
we observe a northwestward geostrophic current, also following the northernmost extension of the high SSH dome generated
by the low SSS plume (Fig. 3i), flowing parallel to the coastline. Acting on the strong mean state meridional temperature
gradient present in the region (Fig. 1a), these shifts in the surface coastal currents lead to changes in horizontal temperature
advection. The differences in horizontal temperature advection (Fig.3m-p) closely resemble those in SST (Fig. 3a-d). Except
for the coastal stripe from 6ºS to 11ºS, freshwater-induced horizontal advection appears to be responsible for the increased
coastal SST in CLIMA. The relative effects of changes in surface currents and in the surface horizontal temperature gradients
present in the area (Fig. 1a) will be investigated further in the next section.

Finally, by calculating the BLT and the squared Brunt-Väisälä frequency, $N^2$, we also assess the freshwater input impacts on
the mean stratification and stability of the water column. BLT and $N^2$ differences among the experiments (Fig. S6) show
extremely similar patterns to the ones observed in SSS (Fig. 3e-h), with a significant negative (positive) SSS difference
implying a significant positive (negative) difference in both BLT and $N^2$ averaged from surface to 50m. It represents an increase
in the water column stratification and a generation of barrier layers at the Congo plume via freshwater discharge in the SETA.
As well as at the Congo River plume, a similar response to the freshening is observed at Kunene River mouth, but with smaller
magnitude (i.e. BLT larger than 8m at 6ºS, and ~3m at 17ºS). Further, the barrier layer induced by a freshwater input does not
fully correspond with the region of increased SST. It is expected that a combination of stronger stratification with a weaker
vertical temperature gradient within the BLs could isolate the surface from the deeper water column layers, reducing the mixing





and contributing to an SST increase. However, it has been shown that this might not always be the case in the SETA due to the counteracting effect of solar radiation penetrating through the very shallow mixed layer (White and Toumi, 2014). However, at the Kunene River mouth the pattern of lower SSS and thick barrier layer seems to boost the SST differences, creating an area of higher SST differences at 17ºS (Fig. 3c).


Overall, we find that including a freshwater discharge in the experiments leads to significant changes in the dynamic and thermodynamic mean state of the SETA. These effects are characterized by the CRD freshwater input reducing the SSS, followed by a halosteric increase in SSH, thereby generating poleward (equatorward) coastal currents south (north) of the river mouth, advecting warmer (cooler) waters to further south (north). Although the observed differences south of the area of the major Congo influence (i.e. south of 13ºS) are small when compared to the river's mouth region (i.e. ~6ºS), they are still significant. Albeit small, the change in the 2000-2018 mean state indicates that the effects presented above are constantly and continuously present. In fact, the isotherms' outcropping latitudes are shifted towards the south in CLIMA compared to NORIV (Fig. S6) as a consequence of this mean state change. Likely due to the constant southward coastal jet, the presence of a freshwater discharge moves the isotherm's position further south. Therefore, the position of the ABF, a front of strong temperature gradient, is also pushed towards the south in the experiment with the freshwater input present. This shift in the ABF location might have substantial impacts on the environmental conditions around this region, which will be further discussed in Sect. 4. In the following section, we detail the warming from advection in the CABA.

### 3.2.2 Advection warming the coastal Angola-Benguela area

In this section, we disentangle the cascade of events described in Sect. 3.2.1 by showing the role of the freshwater-induced dynamical change in increasing the SST at the West African coast. Figure 4 statistically shows the SSH response to the SSS change, the horizontal advection response to the SSH change, and the SST response to the advection change, from the CRMA region to the CABA. Each point corresponds to one of the 228 analyzed months, indicating significant correlations throughout the year, and representing mean state permanent responses.


The difference in SSH in the CRMA is strongly and inversely correlated to the SSS change in the same region (r = -0.81, p < 0.05, Fig. 4a), highlighting the crucial role of the halosteric effect in increasing the SSH at the Congo's mouth. At the same time, a positive and significant correlation is observed between a CRMA SSH change and the horizontal advection in the CABA (Fig. 4b). A 5cm increase in SSH implies a ~6 W/m² increase in horizontal temperature advection in the CABA. Finally, differences in both horizontal temperature advection and SST in the SETA coastal area are also positively and significantly correlated (Fig. 4c), indicating that the stronger advection from CLIMA intensifies the SST compared to NORIV.





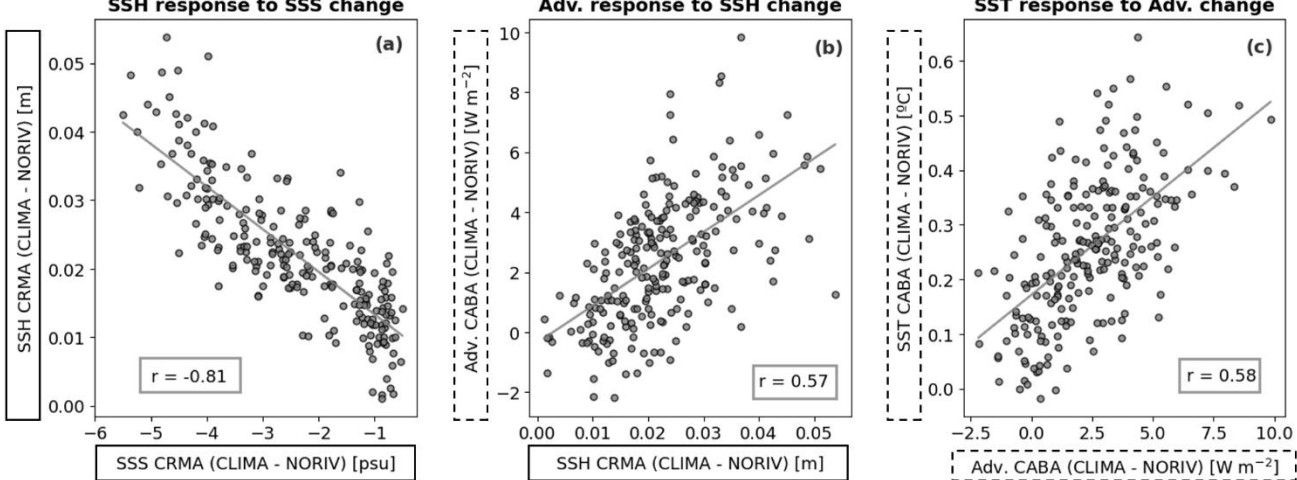

**Figure 4.** Ocean response to land to ocean discharge. **(a)** Linear regression of monthly SSS differences upon monthly SSH differences (CLIMA – NORIV) averaged for CRMA. **(b)** Linear regression of monthly SSH differences upon monthly advection differences (CLIMA – NORIV) averaged for CRMA and CABA, respectively. **(c)** Linear regression of monthly advection differences upon monthly SST differences (CLIMA – NORIV) averaged for CABA area. CRMA and CABA are defined in Fig. 2.

To investigate if the causes of the horizontal temperature advection differences observed between the model experiments stem from changes in the surface currents or from shifts in the horizontal temperature gradients, we calculated the terms by only varying one of these parameters, and then compared the results. Figure S7a depicts total horizontal temperature advection differences [$(-\boldsymbol{v}_{CLIMA} \cdot \nabla T_{CLIMA})$ - $(-\boldsymbol{v}_{NORIV} \cdot \nabla T_{NORIV})$], similar to Fig. 3m-p; Fig. S7b shows the effect by only changing the temperature gradient (i.e. [$(-\boldsymbol{v}_{NORIV} \cdot \nabla T_{CLIMA})$ - $(-\boldsymbol{v}_{NORIV} \cdot \nabla T_{NORIV})$]); and Fig. S7c displays the influence related to changes only in the horizontal velocity (i.e. [$(-\boldsymbol{v}_{CLIMA} \cdot \nabla T_{NORIV})$ - $(-\boldsymbol{v}_{NORIV} \cdot \nabla T_{NORIV})$]). From Fig S7 we can see that by changing only the horizontal velocity in the calculation (Fig. S7c), we generate a spatial pattern and magnitude that mostly resembles the total advection difference (Fig. S7a). The dynamical effect on the advection term generates a similar warming (cooling) signal by the freshwater presence south (north) of the Congo's mouth (i.e. ~6ºS). The difference pattern resulting from only changing the current also depicts a stronger advection extending to ~28ºS, even though the mean state current change at this location is considerably weaker than the strength of this shift at around 6ºS (Fig. 3i). The maximum positive difference in Fig. S7c is located in the ABF region (i.e. 15ºS – 18ºS), supporting the argument that the stronger southward current created by the freshwater input from land pushes the ABF further to the south (Fig. S6).

Finally, it is likely that, by changing the geostrophic dynamics at the West African coast, a freshwater discharge can shift the surface waters distribution in the SETA. For instance, an increased southward transport in the CLIMA experiment implies that warmer and saltier tropical waters push cooler and fresher subtropical waters to further south with the freshwater input. This can be seen not only by the isotherm's outcropping position southward shift in Fig. S6 but also by the increase in SSS south of



17ºS (i.e. a region which is usually not reached by Congo River waters) (Fig. 3). Indeed, both temperature and salinity horizontal gradients present in the SETA (see Figs. 1a and 2a) are important to these southward shifts, however, the strongest differences between the experiments are only significant due to a dynamical change in the surface geostrophic currents. Besides

the advective effects in the CABA, it is still unclear, however, from which mechanisms the freshwater input can generate the coastal warming from 6ºS to 11ºS. In addition, we wonder whether a shift in the surface geostrophic currents might also impact the local geostrophic upwelling. In Sect. 3.2.3 we attempt to answer these questions.

### 3.2.3 Changes in upwelling

In the previous sections, we have shown that the presence of a land to ocean freshwater discharge leads to an intensification

of the surface geostrophic currents in the SETA dominantly in alongshore direction, especially close to the Congo River mouth. The strengthening of the geostrophic dynamics in this region by the freshwater input generates also cross-shore currents. Offshore (onshore) geostrophic flows (Fig. 3i,j) favor coastal upwelling (downwelling). In fact, it has been recently shown that onshore geostrophic flow due to river inflow limits upwelling in the Gulf of Guinea (Alory et al., 2021). To investigate this process at the West African coast, we calculate dynamical upwelling indices for the CLIMA and NORIV experiments.


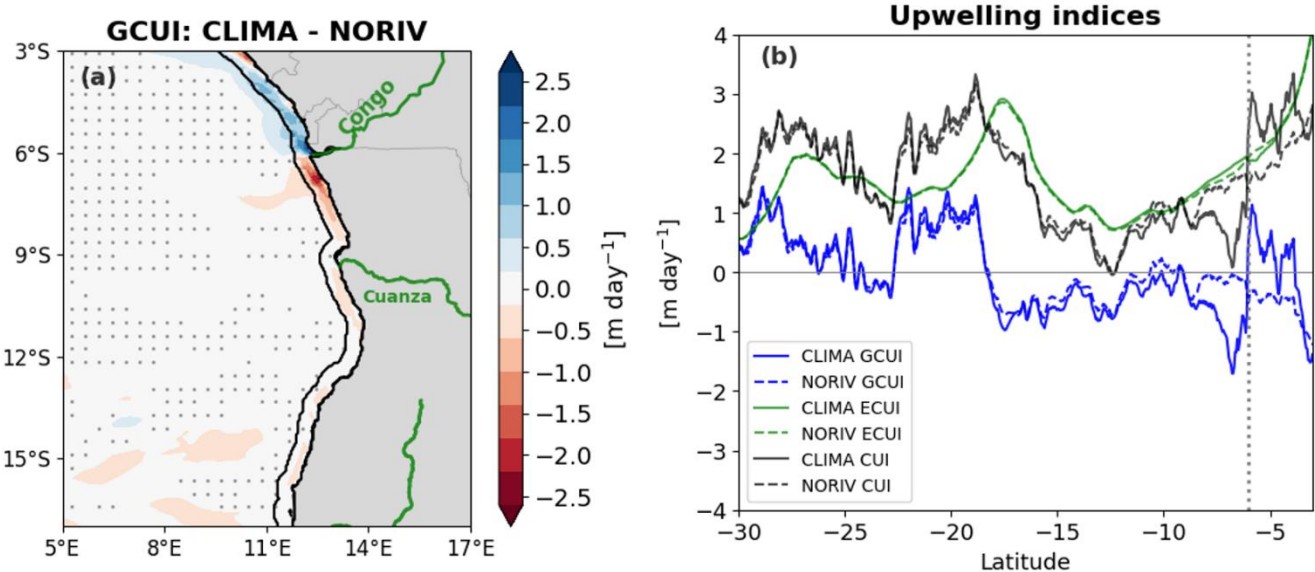

**Figure 5.** Freshwater input impact on upwelling. **(a)** Difference between CLIMA and NORIV for the mean Geostrophic Coastal Upwelling Index (GCUI). Coastal mask in (a) represents 50km away from coast, indicating the region where upwelling occurs (i.e. $L_u$, Eqs. 5, 6). **(b)** Mean upwelling indices averaged for coastal mask depicted in (a). Solid (dashed) lines indicate CLIMA (NORIV) indices. GCUI, ECUI and

CUI are represented by blue, green and black lines, respectively. The pointed line in 6ºS indicates the position of Congo River mouth. Positive (negative) values in both (a) and (b) represent upwelling (downwelling).

Figure 5 shows the mean state differences between CLIMA and NORIV in the calculated upwelling indices. No changes between the experiments were found for ECUIs (Fig. 5b, green lines) along the West African coast. This result was expected



since both experiments are based on an ocean-only model with the same atmospheric wind forcing. Overall, the ECUI
dominates the upwelling intensity (i.e. CUI) along the West African coast, especially in the Benguela Upwelling system, south
of 17ºS (Fig. 5b). In this area, the GCUI is much weaker and close to zero from ~28ºS to 23ºS. It is widely known that wind-
driven upwelling dominates this eastern boundary system (e.g. Bordbar et al., 2021; Brandt et al., 2024; Fennel, 1999). North
of 17ºS, ECUI still dominates but to a lesser extent. An onshore geostrophic flow counteracts the wind-driven upwelling from
17ºS to 10ºS (i.e. negative values for GCUI). It has been recently shown that at this location, in the Angolan upwelling system,
upwelling and high productivity are not governed by the wind forcing, but were shown to be mixing-driven and related to the
passage of Coastally Trapped Waves (CTWs) (Brandt et al., 2024; Körner et al., 2024). Still, from 30ºS to 10ºS no large
differences were observed between the upwelling indices from the different simulations.

However, from 10ºS to further north, and especially around the Congo river mouth region, strong and significant differences
in the GCUI are found between the experiments. From the NORIV experiment, the coastal geostrophic upwelling is nearly
absent around the area of Congo River influence (Fig. 5b). On the other hand, by including land to ocean freshwater input, an
offshore (onshore) geostrophic flow resulting from the low salinity water discharge generates coastal divergence (convergence)
from 6ºS to 4ºS (8ºS to 6ºS) (Fig. 5a,b). The resulting upwelling (downwelling) corresponds to reduced (increased) SST in
CLIMA, when compared to NORIV (Fig. 3a). Thus, the freshwater-induced downwelling might explain the warming signal
from 6ºS to 10ºS, not explained by the strengthened coastal southward advection.

## 4 Conclusions and discussion

In this study, we have shown the effects and mechanisms of the freshwater input presence on the mean state SST in the West
African coastal region. To do that, we focused on comparing two model experiments: one with climatological freshwater
discharge (CLIMA); and one without any discharge from land to ocean (NORIV). The processes are summarized in Fig. 6.
Including a land to ocean freshwater discharge results in a strong reduction in SSS near the coast which generates a halosteric
effect in the water column, increasing the SSH, and creating a geostrophic surface circulation. The halosteric SSH increase as
a result of the CRD creates a primary alongshore SSH gradient that is in balance with a cross-shore geostrophic current, which,
in turn, generates a secondary cross-shore SSH gradient (Fig. 6). While the primary alongshore gradient is associated with the
cross-shore geostrophic current and upwelling (downwelling) north (south) of the Congo River mouth, the secondary gradient
drives the alongshore geostrophic current related to the alongshore advection that reduces (increases) the SST north (south) of
6ºS (Fig. 6). While this signal can propagate southward (Fig. 6) due to the CTW adjustment and affect regions far south, in the
north it must be limited to the region close to the Congo River mouth as there is no equatorward wave propagation. It is also
in agreement with the fact that the SSH maximum is at the coast south of the river's mouth and moves away from the coast
north of it (Fig. 3). In summary, the halosteric increase in SSH produces a southward (northward) coastal geostrophic flow,
with an onshore (offshore) geostrophic component south (north) of 6ºS. The generated southward (northward) coastal jet



advects warmer (cooler) waters to further south (north), from ~10ºS to ~20ºS (~6ºS to ~4ºS). Concomitantly, the onshore (offshore) geostrophic components significantly reduce (increase) the upwelling from 6ºS to 10ºS (6ºS to 4ºS). Hence, the climatological freshwater discharge generated a mean state downwelling (upwelling) and advection that warms (cools) south (north) of the River mouth, leading to a significant increase (decrease) in SST. Furthermore, the southward advection likely

pushes the Angola-Benguela front further south. Overall, it seems that changes in stratification did not play a major role in altering the surface temperatures, especially at the Congo River mouth. In the following, we will discuss the main findings and caveats of our study.



**Figure 6.** Schematic summarizing the processes related to the freshwater input effect on mean state SST at the West African coast. Halosteric effect generating primary alongshore pressure gradient, producing cross-shore geostrophic currents associated with upwelling (downwelling) north (south) of the Congo's mouth, which creates a secondary cross-shore pressure gradient. The secondary gradient drives the alongshore flow responsible for the advection. This signal propagates southward due to CTW adjustment, while in the north it is restricted to region close to river's mouth since there is no equatorward wave propagation. H (L) indicates high (low) pressure area.




The CTRL mean SST presented a warm bias when compared to satellite and reanalysis products. The warm SST bias is indeed an established and longstanding issue in both climate and regional ocean model simulations along the EBUS (e.g. Farneti et al., 2022; Small et al., 2024). Weak upwelling velocities and equatorward surface flow, poor representation of clouds, and model spatial resolution have been discussed as some of the causes of these errors (e.g. Richter, 2015; Bonino et al., 2019; 485  Small et al., 2024). Within the SETA, the Benguela upwelling system appears to be even more challenging to simulate, likely due to the convergence of significantly distinct water masses, the unique spatial structure of the wind field, and its influence on the dynamics of these coastal waters (Bonino et al., 2019; Kurian et al., 2021). For instance, the ABF location and the strength of the AC have been pointed out as major causes of warm biases in general circulation models (Koseki et al., 2018). At the same time, De La Vara et al. (2020) showed that the warm bias in the region is decreased by increasing the oceanic 490  model resolution. Furthermore, Small et al. (2024) recently showed that using the higher resolution JRA55-do as atmospheric forcing in an ocean model also contributes to reducing the SST bias in the Benguela upwelling region, since it can better represent the alongshore winds and its associated downwind surface currents in comparison to the lower resolution CORE, even though a substantial bias relative to observations remains.

Simultaneously, there are also difficulties in well representing SSS fields, especially near the coast and close to river mouths, due to its high variability, strong vertical gradients, and low sampling rates in those regions (Boutin et al., 2021; Martins and Stammer, 2022; Nyadjro et al., 2022). However, both ESACCI and GLORYS12 products have been used and validated in the West African coastal region against independent in situ measurements (Tchipalanga et al., 2018; Martins and Stammer, 2022; Aroucha et al., 2024). Overall, they seem to perform well, with the larger uncertainties located at the Congo mouth, as expected 500  (Martins and Stammer, 2022). In this study, the highest differences between the model SSS and both the satellite and reanalysis products were also found in the SETA coastal region. Further, the direct comparison between CTRL and the ESACCI product could present some caveats, since in this case, we are comparing measurements at different depths (satellite skin layer salinity vs 3m depth as $1^{st}$ level from the model). In regions of strong vertical stratification such as river mouths, the depth level difference between products creates even larger SSS biases, as shown in the Gulf of Guinea by Nyadjro et al. (2022). 505  Furthermore, the uncertainty of JRA55-do Congo River discharge is not well defined and might be substantial. Large discrepancies between this reanalysis forcing and the Brazzaville-Kinshasa gauge measurements have been attributed to the Congo Basin complex hydrology and the lack of field observations of climate variables in the region (Chandanpurkar et al., 2022; Hua et al., 2019). Hence, discrepancies between model and satellite SSS data within this region could be mainly related to the fidelity of these estimates (Chandanpurkar et al., 2022).


Additionally, the SSS variability in the coastal SETA is determined not only by freshwater input from the CRD, but also by the subsequent horizontal advection of the low SSS water by surface currents, with the river plume usually spreading west-northwestward, while some fractions of it being also advected southward along the coast (Houndegnonto et al., 2021; Awo et al., 2022; Ngakala et al., 2023). This southward alongshore advection is subject to interannual variability mainly driven by the





AC and the propagation of CTWs (Awo et al., 2022; Martins and Stammer, 2022). Therefore, the SSS differences between the model simulation and the analyzed datasets are also likely explained by shifts in the freshwater input and more importantly by the surface circulation difference between the three products. In the latter case, the CTRL low SSS from the CRD is confined to further north (i.e. to ~10ºS) than what is observed for both ESACCI and GLORYS (Fig. 2).

Regarding SST variability, patterns of SST anomalies standard deviations from CTRL resemble those in observations (Fig. 2c). Besides the reduced amplitude for extreme events in CTRL, the interannual anomalies are overall well represented. The reason for the reduced SST variability in the model simulation remains uncertain (Fig. S2b-c). Recently, Prigent and Farneti (2024) showed that using JRA55-do atmospheric forcing improved the SST variability simulation in EBUS, including the CABA (see their Fig. 10), when compared to the use of CORE-II forcing. Hence, it is believed that the cause of the variability

underestimation in our case is not in the atmospheric forcing. One possible explanation, as well as for SSS, could be a too weak southward current in CTRL. Similar to the SSS variability, extreme events of SST in the region are also forced by CTW propagation and a southward advection mechanism related to equatorial and local dynamics, usually peaking in boreal spring (e.g. Aroucha et al., 2024; Bachèlery et al., 2020; Imbol Koungue et al., 2019; Rouault et al., 2007, 2018; Florenchie et al., 2004). Therefore, a weaker representation of the AC might generate reduced SST anomalies during extreme events. In fact,

Benguela Niños for which southward advection played an important forcing role, such as 2001 and 2011 (Rouault et al., 2007, 2018), were shown to have reduced SST anomalies in CTRL (Fig. 1c). At the same time, the January-May alongshore southward current is underestimated by the model in relation to the mooring measurements (Fig. S3). This fact also implies the reduced SSS variability in the CABA in CTRL, another likely reason for the underestimated CTRL SST variability, since low SSS waters have been linked to extreme positive SST events in the region (Aroucha et al., 2024; Lübbecke et al., 2019).


In this study, we focused on the climatological runoff impact on the SST mean state (i.e. CLIMA vs. NORIV), since no significant differences in the SST means were observed between the simulations with climatological vs. interannually varying runoff (i.e. CTRL vs. CLIMA, Fig. S4). This does not necessarily imply, however, that an interannually varying runoff does not influence the SST variability in the region. It has been recently shown through observational datasets that a freshwater

input in the ABA could indeed boost extreme warm events in this area through increasing stratification and reducing vertical mixing (Aroucha et al., 2024; Lübbecke et al., 2019). For this, a combination of anomalously high CRD with stronger southward advection and the passage of a CTW is required to bring the low SSS waters close to the ABF region (Martins and Stammer, 2022). In the 19 simulated years (2000-2018), however, the southernmost extension of a coastal SSS difference between CTRL and CLIMA was ~14ºS (-0.1 psu in December 2010, not shown), still too far north for the expected influence

to take place. Hence, from the experiments analyzed here, not much can be said regarding CRD anomalies influencing SST variability in the ABA. Nevertheless, relevant outcomes regarding the freshwater influence on the SST mean state in the SETA can be concluded from the present study.





During the last few years, modeling studies have been addressing the processes by which a freshwater discharge could impact
SSTs. The main argument is that increasing the input of low salinity waters into the ocean would increase SSTs by
strengthening the stratification and inhibiting the upwelling of cold waters (e.g. Topé et al., 2023; Zhang and Busalacchi,
2009). On the other hand, at the Congo River mouth, the work from White and Toumi (2014) pointed to no significant influence
of an increased stratification by freshwater in warming the sea surface. Instead, they showed that the mixed layer shoaling due
to the Congo discharge generated a considerable heat loss to beneath this thin layer, which then exceeded the reduced vertical
mixing impact, even though barrier layers were formed (White and Toumi, 2014). In fact, this latter result agrees with what
we found in this study. In spite of the significant barrier layer difference between CLIMA and NORIV at the river plume, the
barrier layer generation area differs from the observed spatial warming pattern (solely concentrated at the coast, south of 6ºS).
We hypothesize that this is due to the same argument presented by White and Toumi (2014); i.e. the surface heat loss due to
the very shallow MLD at the river mouth. Additionally, their simulations demonstrated a significant coastal warming impact
south of 6ºS (White and Toumi, 2014). Although the authors did not explore this further, they suggested that a change in ocean
dynamics due to the river's presence is responsible for the observed coastal warming pattern. Here we dug into this mechanism,
showing that indeed the coastal warming was a consequence of modified ocean dynamics via a change in geostrophy.
Furthermore, we showed that those changes in SST are restricted to the coast since the geostrophic current strengthening
reflects the location of the coastal freshwater input.


It is long known that river discharges can impact oceanic SSH (e.g., Meade and Emery, 1971; Piecuch et al., 2018;
Chandanpurkar et al., 2022). To the best of our knowledge, however, this is the first study to report that a halosteric change in
SSH and its gradients due to a freshwater discharge can impact the mean state coastal SST at an EBUS, via a change in
geostrophic currents, subsequently altered horizontal temperature advection, and upwelling. Alory et al. (2021) found
weakened upwelling in the Gulf of Guinea by an onshore geostrophic current due to the River Niger input, also pointing to the
possibility of a similar effect next to other large river plumes. The lateral salinity-generated pressure gradient at river plumes
can induce surface geostrophic flows projecting at the coast (Fong and Geyer, 2002). Here we show that the Congo River can
not only limit upwelling to the south of its estuary by this generated onshore geostrophic current, but also can induce upwelling
north of the river plume via an offshore component of this geostrophic flow (Fig. 5). This work highlights the importance of
properly understanding the impact of high freshwater input in ocean mixing and stratification, as well as in the coastal
dynamics. The results presented here might become even more relevant in the future when considering recent studies showing
upwelling shifts at the West African coast due to wind-related geostrophic deviations in climate change scenarios (Ayissi et
al., 2024; Jing et al., 2023). The role of geostrophic flows in EBUS mean state and long-term changes is more prominent in
the Atlantic basin when compared to the Pacific (Jing et al., 2023). In addition, Jing et al. (2023) recall an ongoing discussion
of the interaction between upwelling and other greenhouse warming-related processes (e.g. stratification, mesoscale activity)
in these regions. As river runoffs are expected to be amplified in the future (Müller et al., 2024; Aloysius and Saiers, 2017),
halosteric-related shifts in geostrophic flows can appear as an additional mechanism in this debate.



Furthermore, as we showed that the freshwater presence pushes the ABF southward, we wonder if the future runoff
amplification could also contribute to the recently observed trend of warming off the Angolan coast, where fresher and warmer
tropical waters are moving poleward due to an AC intensification (Roch et al., 2021; Tomety et al., 2024). Finally, the
mechanism shown here could also play a role in Benguela Niño events in this area. Most of the advective warming from the
southward geostrophic current is located within the CABA where these events occur. Hence, in years of increased freshwater
discharge, these warming events could be boosted by different mechanisms: the reduced vertical mixing via increased
stratification (Aroucha et al., 2024), and the strengthening of the southward advection and downwelling due to the halosteric
SSH increase, as shown in this work.

*Code and data availability.* The data and scripts that support the findings of this study are available through GEOMAR at
https://hdl.handle.net/20.500.12085/2b927bcd-afab-4bc6-ba97-634d09435daa (Aroucha and Schwarzkopf, 2024). .
Experiments identifiers are INALT20.L46-KFS10X (CTRL), INALT20.L46-KFS111 (CLIMA), and INALT20.L46-KFS106
(NORIV). Other datasets used in this work are publicly available under the following links: OISST
(https://psl.noaa.gov/data/gridded/data.noaa.oisst.v2.highres.html); GLORYS12
(https://data.marine.copernicus.eu/product/GLOBAL_MULTIYEAR_PHY_001_030/download); ESACCI
(https://data.ceda.ac.uk/neodc/esacci/sea_surface_salinity/data/v03.21/30days); Congo River discharge (https://hybam.obs-
mip.fr/data/); Current velocities from mooring data (https://doi.org/10.1594/PANGAEA.870917;
https://doi.pangaea.de/10.1594/PANGAEA.909911; https://doi.org/10.1594/PANGAEA.909913;
https://doi.pangaea.de/10.1594/PANGAEA.939249; https://doi.org/10.1594/PANGAEA.962193); GLOCURRENT data
(https://data.marine.copernicus.eu/product/MULTIOBS_GLO_PHY_MYNRT_015_003/download?dataset=cmems_obs_mo
b_glo_phy-cur_my_0.25deg_P1M-m_202311).

*Author Contribution.* LCA and JFL took part in conceptualizing and outlining the paper. LCA wrote the first draft and produced
the figures. FUS and AB developed and ran the sensitivity experiments. All authors contributed to the discussion of the results
and the writing of the manuscript.

*Competing Interests.* The authors declare that they have no conflict of interest.

*Acknowledgments.* LCA acknowledges the German Academic Exchange Service (DAAD). The model integrations were
enabled by the provision of computing resources on the high-performance computing system JUWELS at the Jülich
Supercomputing Centre (JSC) in the framework of the Earth System Modelling Project (ESM) and at the North German
Supercomputing Alliance (HLRN).



*Financial Support.* This study was supported by the German Academic Exchange Service (DAAD) via the Doctoral Research Grant 57552340, and by the Bundesministerium für Bildung und Forschung (grant no. 03F0796A (SPACES-II-CASISAC)).

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
