# Peer review of "River discharge impacts coastal Southeastern Tropical Atlantic sea surface temperature and circulation: a model-based analysis"

_EGUsphere, 2024_

## Referee Comment (RC1)

Review of "River discharge impacts coastal Southeastern tropical Atlantic sea surface temperature and circulation: a model-based analysis" by Aroucha et al. 2024

The study uses high-resolution model sensitivity experiments to understand the impact of freshwater input on sea surface temperature (SST) variability in the southeastern tropical Atlantic Ocean. Results from model sensitivity runs with climatological freshwater forcing and no freshwater forcing suggest that freshwater-induced SST warming occurs to the south and cooling to the north of the Congo River mouth. The authors propose that freshwater discharge from the Congo River causes halosteric changes in sea surface height, which results in alongshore downwelling circulation and leads to advective SST warming along the coast south of the river mouth. Similarly, the low salinity-induced alongshore circulation to the north of the river mouth is associated with upwelling and cooling of SST. Furthermore, the implications of the southward advection of river water on coastal upwelling are discussed.

This is an important study that highlights the impact of Congo River discharge and low salinity on coastal ocean circulation dynamics, SST variability, and coastal upwelling. The paper is well-organized, with good-quality figures. The manuscript may be considered for publication after the authors have addressed the major and minor comments listed below.

Major comments:

1. Figures 1 and 2: While I appreciate the thorough validation of the model data with in situ observations and the reanalysis dataset, the timeseries plots in both figures appear very cluttered, and the different colored curves are difficult to distinguish. My suggestion would be to remove the NO RIV' and 'CLIM' curves from these figures and make a new figure with SST and SSS from CTL, NO RIV and CLIM runs, if possible.

2. Figure 3: The velocity vectors are not clearly visible in panels i-l. Is it possible to increase the arrow length? According to the proposed mechanism (Fig. 6), one would expect to see negative SSH differences to the north of the Congo River mouth which depicts upwelling associated with advective low SST values. But Fig. 3(i) shows positive SSH values all along the coast. Can this be explained?

3. It is surprising to see that there is no difference in SSS between the CTL and CLIM runs. Interannual variability in SSS is known to be tightly linked to the interannual variability in Congo River discharge. However, the model discharge does not seem to align well with the observed discharge values at the Kinshasa station (Fig. S1). Do you think the discrepancies in model runoff forcing could be a possible reason for this?

4. Are these linear regression plots and reported correlation values calculated at zero lag? I would expect there to be a lag of 1-2 months between the SSS at the Congo River mouth and the SSH/SST in the coastal Angola-Benguela area, as it takes time for the river water to be advected south along the coast. Can you check this by plotting lagged correlation between the variables?

5. Fig. 4: It might be useful to add a panel showing the linear regression plot between SSS in CRMA and SST in CABA.

6. Would it be possible to show the horizontal advection in °C/day instead of W/m² for easier comparison with the SST plots?

7. Fig. 5 shows that the Ekman upwelling index contributes significantly more to coastal upwelling than the Geostrophic upwelling index. Additionally, there is little difference between the CLIM and NO RIV runs for the ECUI. This figure does not seem to add much to the discussion and could be moved to the supplementary material. Instead, I recommend moving Fig. S7 to the main article, as it illustrates the contribution of geostrophic currents to horizontal temperature advection. This is just a suggestion; ultimately, it is up to the authors to decide.

Minor comments:

1. There are too many acronyms (e.g., SETA, CTW, CRMA, CABA, CUI, ECUI, GCUI, etc.), which can make it difficult for the reader to follow. Please consider reducing the number of acronyms to improve clarity and simplicity.

2. Line 10: Suggest adding "significant" freshwater input from land.

3. Lines 31-32: Please mention the longitude of the mouth of the rivers as well.

4. Line 174: Why were the horizontal advection values within 20 km distance neglected? Is it because of large errors? Add a sentence explaining that.

5. Line 249-250: It is not clear what weak SSS variability in CTL run means here. I see significant variability in CTL run SSS in Fig. 2d.

6. Fig. 6: You might want to say what the blue dotted arrow represents in the right-side graphic legend.

7. Line 439: The negative values of GCUI seem to extend from 17S to 6S.

---

## Author Comment (AC1)

**RESPONSE TO REVIEWER'S COMMENT (RC1)**

for the manuscript "*River discharge impacts coastal Southeastern Tropical Atlantic sea surface temperature and circulation: a model-based analysis*" by Aroucha et al., submitted to *Ocean Science.*

*We thank the reviewers for their thorough evaluation of our manuscript and their suggestions. Below, we provide a detailed response to all reviewer's queries.*

*To address their comments, we have revised all six figures in the main manuscript and added two new figures to the Supplemental Material. Consequently, the order and numbering of supplementary figures might also have changed. In the "TrackedChanges" file you can see all the modifications made and the "Main" file represents the final revised paper with the changes inserted.*

*References cited in this document are included at the end. Responses to individual comments are provided below, with specific references to the corresponding lines and sections in the revised manuscript. For clarity, our responses are highlighted in blue font throughout this response letter.*

**Anonymous Referee #1 (RC1)**

Review of "River discharge impacts coastal Southeastern tropical Atlantic sea surface temperature and circulation: a model-based analysis" by Aroucha et al. 2024

The study uses high-resolution model sensitivity experiments to understand the impact of freshwater input on sea surface temperature (SST) variability in the southeastern tropical Atlantic Ocean. Results from model sensitivity runs with climatological freshwater forcing and no freshwater forcing suggest that freshwater-induced SST warming occurs to the south and cooling to the north of the Congo River mouth. The authors propose that freshwater discharge from the Congo River causes halosteric changes in sea surface height, which results in alongshore downwelling circulation and leads to advective SST warming along the coast south of the river mouth. Similarly, the low salinity-induced alongshore circulation to the north of the river mouth is associated with upwelling and cooling of SST. Furthermore, the implications of the southward advection of river water on coastal upwelling are discussed.

This is an important study that highlights the impact of Congo River discharge and low salinity on coastal ocean circulation dynamics, SST variability, and coastal upwelling. The paper is well-organized, with good-quality figures. The manuscript may be considered for publication after the authors have addressed the major and minor comments listed below.

Major comments:
1. Figures 1 and 2: While I appreciate the thorough validation of the model data with in situ observations and the reanalysis dataset, the timeseries plots in both figures appear very

cluttered, and the different colored curves are difficult to distinguish. My suggestion would be to remove the NO RIV' and 'CLIM' curves from these figures and make a new figure with SST and SSS from CTL, NO RIV and CLIM runs, if possible.

R. Thank you for pointing this out. We removed from both Figures 1(d-e) and 2(d-g) the NORIV and CLIM curves. Now Figures 1 and 2 depict only the difference between the reanalysis and satellite products for both SST and SSS, respectively, compared to the CTRL run. We also added new Figures S2 and S3 with the curves for the three simulations (CTRL, CLIMA, and NORIV) for a better comparison among them. Please find the new figures in the Supplemental Material.

2. Figure 3: The velocity vectors are not clearly visible in panels i-l. Is it possible to increase the arrow length? According to the proposed mechanism (Fig. 6), one would expect to see negative SSH differences to the north of the Congo River mouth which depicts upwelling associated with advective low SST values. But Fig. 3(i) shows positive SSH values all along the coast. Can this be explained?

R. First, thanks for pointing out this issue in Figure 3. We now increased the arrow length and size in Figure 3(i-l) for a better visualization. Please note that the vector scale also changed. Second, thanks as well for your question. The proposed mechanism in Figure 6 describes a primary meridional pressure gradient north and south of the river mouth, which means that the SSH increase directly at the river mouth is higher than north and south of it. However, since these changes refer to differences between runs with and without the river discharge, the SSH difference around the river will always be positive. Due to the spatial difference in the SSH magnitude shift from one experiment to the other (CLIMA – NORIV), the pressure gradients responsible for changing the coastal dynamics are present. Please see below Figure R1, depicting a primary negative meridional pressure gradient north of the river mouth and a positive meridional pressure gradient south of it. Although there are these pressure gradients from the Congo mouth toward north and south, all SSH values are positive (see inverted x-axis in Figure R1).

[Figure]

**Figure R1** *SSH differences between CLIMA and NORIV mean states (CLIMA-NORIV). Mean 200km off coast (x-axis inverted). H stands for higher pressure and L for lower pressure.*

3. It is surprising to see that there is no difference in SSS between the CTL and CLIM runs. Interannual variability in SSS is known to be tightly linked to the interannual variability in Congo River discharge. However, the model discharge does not seem to align well with the observed discharge values at the Kinshasa station (Fig. S1). Do you think the discrepancies in model runoff forcing could be a possible reason for this?

R. Thank you for the question. Yes, the discrepancies between model runoff forcing and the Brazzaville station could be a reason for the almost identical SSS interannual variability between CTRL and CLIM. These discrepancies have been attributed to the complex hydrology and the lack of observational data in the Congo basin (Chandanpurkar et al., 2022; Hua et al., 2019). In fact, Chandanpurkar et al. (2022) argue that the bi-modal rainfall distribution over this river basin due to the poorly understood thick rainforest interaction with hydrology combined with the seasonal shift of the Intertropical Convergence Zone are some reasons for such differences in the discharge. This then limits the JRA55 atmospheric reanalysis performance over the Congo basin. We discuss this point in Sect. 4 "Conclusions and discussion", from L.517 to L.522.

4. Are these linear regression plots and reported correlation values calculated at zero lag? I would expect there to be a lag of 1-2 months between the SSS at the Congo River mouth and the SSH/SST in the coastal Angola-Benguela area, as it takes time for the river water to be advected south along the coast. Can you check this by plotting lagged correlation between the variables?

R. Thank you for your question. Yes, the linear regression plots and correlation values are at zero lag. Please see below Figure R2 with the lagged correlation between the variables presented in Figure 4 of the paper. In fact, the highest correlation values observed are at zero correlation lag (Figure R2). Indeed, considering a particle from the Congo's mouth at 6ºS being transported to the south by a southward coastal current of 0.2m/s (Fig. 3), this particle would be advected around 4-5 degrees of latitude, reaching ~10º - 11ºS within a month, which agrees with our zero-lag correlation between an SSH change at Congo's mouth and an SST change at Angola-Namibia coast. It is important to note that interpreting a lag correlation of the differences between the two experiments is not straightforward, as the processes evolve differently in each simulation also because of different mean states. Therefore, it is challenging to determine a specific timescale for linking the processes shown in Figure 3. Still, the linear regression indicates that changes in SSS and SSH due to the freshwater input are simultaneously associated with an advection response and consequently a change in SST.

[Figure]

***Figure R2*** *-Ocean response to land-to-ocean discharge (lead-lag correlations). (a) Linear regression of monthly SSS differences upon monthly SSH differences (CLIMA – NORIV) averaged for CRMA. (b) Linear regression of monthly SSH differences upon monthly advection differences (CLIMA – NORIV) averaged for CRMA and CABA, respectively. (c) Linear regression of monthly advection differences upon monthly SST differences (CLIMA – NORIV) averaged for CABA area.*

5. Fig. 4: It might be useful to add a panel showing the linear regression plot between SSS in CRMA and SST in CABA.

R. Thank you for your suggestion. We show this regression below (Fig. R3c) but did not add the panel in the manuscript since the correlation is not significant. Since there are several processes involved in this SST change associated with a SSS change (e.g. halosteric effect of increasing SSH, changing mixed layer depth and mixed layer heat budget, coastal current generation, advection, etc.) we believe that showing the step-by-step correlation for each process highlights and delineates in a better way the processes related to a freshwater input impact on the SST changes at the southwestern African coastal fringe than the direct regression of SSS change at CRMA to CABA SST shifts. Still, the CABA SST response to SSH change at CRMA (Fig. R3a) and CABA advection response to the SSS change at CRMA (Fig. R3b) are significantly correlated.

[Figure]

***Figure R3*** *-Ocean response to land-to-ocean discharge. (a) Linear regression of monthly SSH differences upon monthly SST differences (CLIMA – NORIV) averaged for CRMA. (b) Linear regression of monthly SSS differences upon monthly advection differences (CLIMA – NORIV) averaged for CRMA and CABA, respectively. (c) Linear regression of monthly SSS differences upon monthly SST differences (CLIMA – NORIV) averaged for CABA area.*

6. Would it be possible to show the horizontal advection in °C/day instead of W/m² for easier comparison with the SST plots?

R. Thank you for the suggestion! Yes, it would be possible. The horizontal advection term is now in units of ºC/day in all plots (please see Figs. 3, 4 and S9). For this, we also needed to change Eq. 4 in Section 2.4, L.171, since the units are now independent of the seawater density, specific heat capacity and mixed-layer depth.

7. Fig. 5 shows that the Ekman upwelling index contributes significantly more to coastal upwelling than the Geostrophic upwelling index. Additionally, there is little difference between the CLIM and NO RIV runs for the ECUI. This figure does not seem to add much to the discussion and could be moved to the supplementary material. Instead, I recommend moving Fig. S7 to the main article, as it illustrates the contribution of geostrophic currents to horizontal temperature advection. This is just a suggestion; ultimately, it is up to the authors to decide.

R. Thank you for your suggestion! Indeed, the ECUI does not significantly change from one simulation to the other. This is explained since the atmospheric forcing is the same for both runs. However, the purpose of this figure is to depict changes specifically in the geostrophic upwelling around the Congo's mouth. Although the ECUI dominates the coastal upwelling in relation to the GCUI, we believe that the significant shift in GCUI in the simulations including the river discharge is of great importance. The downwelling and upwelling south and north of 6ºS, respectively, associated with a change in the geostrophic coastal currents due to the freshwater input explain the observed coastal SST difference from 6ºS to 11ºS. This shift in GCUI also contributes significantly to changing the total upwelling around the Congo's mouth. Therefore, we opted to keep Fig.5 in the manuscript, though we appreciate your suggestion.

Minor comments:

1. There are too many acronyms (e.g., SETA, CTW, CRMA, CABA, CUI, ECUI, GCUI, etc.), which can make it difficult for the reader to follow. Please consider reducing the number of acronyms to improve clarity and simplicity.

R. Thanks for pointing that out. We removed the acronyms for CUI (total upwelling indice), AC (Angola Current), CTW (Coastally Trapped Waves), and EBUS (Eastern Boundary Upwelling Systems). Please see these changes throughout the text. Unfortunately, we believe some acronyms are still necessary to define a specific area (e.g. CRMA, CABA) or index (GCUI, ECUI). Still, we hope these changes can already improve clarity for the reader.

2. Line 10: Suggest adding "significant" freshwater input from land.
R. Thank you for the suggestion. The term is now added in L.10 in the Abstract.

3. Lines 31-32: Please mention the longitude of the mouth of the rivers as well.
R. Thank you for observing this. The river's mouth both latitudes and longitudes are now included in the text in L.34.

4. Line 174: Why were the horizontal advection values within 20 km distance neglected? Is it

because of large errors? Add a sentence explaining that.

R. Thanks for the question. Yes, the horizontal advection values within the 20km distance to the coast are neglected due to the large error close to the coast associated with the horizontal temperature gradient calculation. Please see below Fig. R4. how it looks without neglecting the 20km off the coast. We have now included an explaining sentence in Section 2.4 at the main text in L.175.

[Figure]

*Figure R4* - *Differences between CLIMA and NORIV mean states (CLIMA-NORIV) for Horizontal Advection (m-p) neglecting the values 20km off the coast (left) and including those values (right). Stippled grey areas indicate where the difference is not significant in a 95% confidence level.*

5. Line 249-250: It is not clear what weak SSS variability in CTL run means here. I see significant variability in CTL run SSS in Fig. 2d.

R. Thank you for the comment. Indeed, the CTRL SSS variability is significant, as shown in Figure 2d. However, here we meant that the CTRL SSS variability in Coastal Angola-Benguela Area (CABA) is weaker than the SSS variability observed in both satellite (ESACCI) and reanalysis (GLORYS) products (new Fig. S4d-f). We added this information to the sentence from L.259 to L.263; hopefully, it reads better now.

6. Fig. 6: You might want to say what the blue dotted arrow represents in the right-side graphic legend.

R. Thanks for pointing this out. The blue dotted arrow also represents a geostrophic current related to the primary pressure gradient. We have now included this information in the legend of Fig. 6.

7. Line 439: The negative values of GCUI seem to extend from 17S to 6S.

R. Thanks for catching this. Indeed, the values extend to 6ºS. We replaced "17ºS to 10ºS" by "17ºS to 6ºS", and it can be found in L.447.

**References cited in this document**

Chandanpurkar, H. A., Lee, T., Wang, X., Zhang, H., Fournier, S., Fenty, I., Fukumori, I., Menemenlis, D., Piecuch, C. G., Reager, J. T., Wang, O., and Worden, J.: Influence of Nonseasonal River Discharge on Sea Surface Salinity and Height, J Adv Model Earth Syst, 14, e2021MS002715, https://doi.org/10.1029/2021MS002715, 2022.

Hua, W., Zhou, L., Nicholson, S. E., Chen, H., and Qin, M.: Assessing reanalysis data for understanding rainfall climatology and variability over Central Equatorial Africa, Clim Dyn, 53, 651–669, https://doi.org/10.1007/s00382-018-04604-0, 2019.

---

## Author Comment (AC2)

**RESPONSE TO REVIEWER'S COMMENT (RC2)**

for the manuscript "*River discharge impacts coastal Southeastern Tropical Atlantic sea surface temperature and circulation: a model-based analysis*" by Aroucha et al., submitted to *Ocean Science.*

*We thank the reviewers for their thorough evaluation of our manuscript and their suggestions. Below, we provide a detailed response to all reviewer's queries.*

*To address their comments, we have revised all six figures in the main manuscript and added two new figures to the Supplemental Material. Consequently, the order and numbering of supplementary figures might also have changed. In the "TrackedChanges" file you can see all the modifications made and the "Main" file represents the final revised paper with the changes inserted.*

*References cited in this document are included at the end. Responses to individual comments are provided below, with specific references to the corresponding lines and sections in the revised manuscript. For clarity, our responses are highlighted in blue font throughout this response letter.*

**Anonymous Referee #2 (RC2)**

Review of "River discharge impacts coastal Southeastern Tropical Atlantic sea surface temperature and circulation: a model-based analysis"

**General comments**
Understanding sea surface temperature (SST) and circulation in the southeastern tropical Atlantic is crucial for understanding upwelling dynamics, air-sea interactions, and other related processes. However, the limited availability of in-situ observations in this region has hindered a comprehensive understanding of these dynamics. The objectives of this study are therefore both timely and commendable. This study investigated the impact of river discharge on the mean state SST in coastal western Africa. Through modeling efforts, several scenarios and experiments were conducted to analyze the influence of large river outflows on SST and geostrophic flow in the region. The results indicate that river outflows generate a halosteric effect in the water column, leading to an increase in sea surface height (SSH) and inducing geostrophic circulation in the surface ocean. The resulting SSH gradient drives upwelling and downwelling processes, along with alongshore advection, which collectively alter SST. The paper is well-written, and the analyses are comprehensive, effectively addressing the research questions. However, the following recommendations should be considered for revision prior to acceptance and publication

**Specific comments**
1. L57: does the barrier layer not strengthen the vertical temperature gradient, hence reducing the impact of vertical mixing?
R. Thanks for the question. The barrier layer weakens the vertical temperature gradient between the mixed-layer and the waters below it. With a barrier layer (Figure R1 below, right plot), the mixed-layer (in this situation defined by a change in density rather than a temperature change) sits within the isothermal layer depth. Therefore, since the temperatures of waters within the isothermal layer are almost constant, the vertical temperature gradient is weakened in such cases.

The weakening of the vertical temperature (dT/dz) is then proportional to a reduction in the turbulent heat flux below the mixed-layer ($J_h$), which is defined as $J_H = - dT/dz*K_\rho$ (e.g. Hummels et al. (2020) and Körner et al. (2023)).

[Figure]

**Figure R1**. *A schematic view of a river-induced BL: the mixed layer is shallowed, whilst the top of the thermocline remains constant. Horizontal lines depict the bottom of the density and temperature mixed layers. Quantitative values and profile shapes are for illustration only. From White and Toumi (2014), Figure 8(a).*

2. L62: why averaging from surface to 50m depth?

R. We believe this comment refers to L.162 instead of L.62. Thanks for the question. We average the squared Brunt-Väisälä frequency values from the surface to 50m to depict a spatial view of the changes in stratification in the SETA. Since the MLD in this region is usually shallower than 50m (Körner et al., 2023; Aroucha et al., 2024), we believe that averaging until this depth well-captures the stratification shifts generated by the freshwater discharge in a 2D field. We have added this information in the main text, from L.164 to L.166.

3. L201-225 and Figure 1. It may be helpful to the reader to include the reasons for the biases here. Mainly, what account for the differences? Comment on the fact that the satellite product measures skin temperature while for the model, "surface" temperature is at 3m.

R. Thanks for pointing that out! From paragraphs 2 to 5 in Sect. 4 (Conclusions and Discussion) we extensively discuss the observed biases not only in the SST and SSS model data mean state, but also in its variability and the U, V, and CRD fields. We indeed mention the skin measurements compared against the 3m model "surface" temperature, but only in terms of SSS. This comment is now added also to the SST biases discussion in paragraph 2 of Sect. 4, from L.504 to L.506.

4. L304: The vectors on these plots are hard to see. Please refine.

R. Thank you! Following what was pointed out by Reviewer #1 in their Major Comment 2, we increased the arrow length and size, also changing the vector scale. We hope that the vectors in the new Figure 3 are now clearly visible.

Technical corrections

1.  L31: The "West African" description is a bit confusing. Yes, its on the western part of Africa, but "West Africa" typically refers to the geo-political description.

R. Thank you for this helpful comment. Indeed, West Africa seems to be referring to the geopolitical description. To avoid this and be more precise in defining our area, we now refer to our region of interest as the southwestern African coast. You can find this new denomination throughout the paper.

2.  L122: spell out NOAA

R. Done. You can find NOAA spelled out in L.126-127.

3.  L127: "of bias-corrected SSS"

R. Thanks! Now corrected. See L. 131.

4.  L132: the spell out of NEMO should come earlier, in L91

R. Indeed! Thank you. You can find NEMO spelled out in L.96-97 now.

5.  L139: Congo River discharge has earlier been abbreviated to CRD. Please maintain consistency.

R. Thank you for pointing that out. Now Congo River discharge is abbreviated in L.36 and referred to as CRD throughout the text.

**References cited in this document**

Aroucha, L. C., Lübbecke, J. F., Körner, M., Imbol Koungue, R. A., and Awo, F. M.: The Influence of Freshwater Input on the Evolution of the 1995 Benguela Niño, JGR Oceans, 129, e2023JC020241, https://doi.org/10.1029/2023JC020241, 2024.

Hummels, R., Dengler, M., Rath, W. et al. Surface cooling caused by rare but intense near-inertial wave induced mixing in the tropical Atlantic. Nat Commun 11, 3829 (2020). https://doi.org/10.1038/s41467-020-17601-x

Körner, M., Brandt, P., and Dengler, M.: Seasonal cycle of sea surface temperature in the tropical Angolan Upwelling System, Ocean Sci., 19, 121–139, https://doi.org/10.5194/os-19-121-2023, 2023.

White, R. H. and Toumi, R.: River flow and ocean temperatures: The Congo River, J. Geophys. Res. Oceans, 119, 2501–2517, https://doi.org/10.1002/2014JC009836, 2014.

---

## Author Comment (AC3)

**RESPONSE TO EDITOR'S COMMENT**

for the manuscript "*River discharge impacts coastal Southeastern Tropical Atlantic sea surface temperature and circulation: a model-based analysis*" by Aroucha et al., submitted to *Ocean Science.*

*We thank the editor for her thorough evaluation of our manuscript and suggestions. Below, we provide a response to the editor's comment.*

*References cited in this document are included at the end. Responses to individual comments are provided below, with specific references to the corresponding lines and sections in the revised manuscript. For clarity, our responses are highlighted in blue font throughout this response letter.*

**Editor's comment (EC1)**

This is an interesting model experiment paper. There are a couple of things I spotted on initial review that I'd like you to address when you come to revision - there are some units psu for salinity, which should be removed as there are no units for salinity on the practical salinity scale. And there are a few instances of referring to something being 'in' a reference which should be replaced with 'by'. Personally, I don't like the use of parentheses for opposites, as you have in the abstract. Since there is no word limit for Ocean Science, it would be clearer to write the two cases out in full. But I know plenty of people use this formulation. Thanks for submitting your work to Ocean Science.

R. Thank you for the comments and suggestions. We have removed the salinity units in the manuscript and the figure's labels. We also revised our abstract and conclusion paragraph in Sect. 4 to replace the use of parentheses with a full write-out of each case. Finally, we fully revised the manuscript writing.

---

## Author Comment (AC4)

**RESPONSE TO COMMUNITY'S COMMENT**

for the manuscript "*River discharge impacts coastal Southeastern Tropical Atlantic sea surface temperature and circulation: a model-based analysis*" by Aroucha et al., submitted to *Ocean Science.*

*We thank the community for its interest in our manuscript and suggestions. Below, we provide a response to the community's comments.*

*References cited in this document are included at the end. Responses to individual comments are provided below, with specific references to the corresponding lines and sections in the revised manuscript. For clarity, our responses are highlighted in* blue font *throughout this response letter.*

**Community comment (CC1):**

I congratulate the authors for this very interesting work.

However I note that our Alory et al. (2021) paper is cited several times (L70, L423, L570) but its results are misinterpreted. It is true that our work hypothesis was that the Niger River could limit coastal upwelling in the Gulf of Guinea through the generation of an onshore geostrophic flow, But in the end, we found that this onshore geostrophic flow already existed in our simulation without river. While there was an additional onshore geostrophic flow due to halosteric effects when adding the Niger River in the model, as you find south of the Congo River, we found that it did not significantly affect our GCUI index as the increased uG was compensated by a reduced MLD. Please correct your interpretation when citing our paper.

This also leads to the following question:

When computing the GCUI (L180) in simulations with/without river, did you take into account the changes in MLD?

R. Thanks for the comment. We apologize for the misinterpretation. Indeed, your paper highlights the river-induced mixed-layer thinning compensation effect on the current change. We reviewed our interpretation when citing the Alory et al. (2021) paper throughout the manuscript. Please see these changes in the main final manuscript from L.73-75, L.428-431, and L.585-588. Regarding the GCUI computation, we do take into account the changes in the MLD, as now described in L.200-202. However, the MLD changes in simulations with/without river at the 50km coastal band (i.e. the here defined cross-shore width where upwelling occurs) are less significant than the changes in the coastal current within the same coastal region. Hence, a significant change in the GCUI around the Congo's mouth is observed when comparing both simulations.

**References cited in this document**

Alory, G., Da-Allada, C. Y., Djakouré, S., Dadou, I., Jouanno, J., and Loemba, D. P.: Coastal Upwelling Limitation by Onshore Geostrophic Flow in the Gulf of Guinea Around the Niger River Plume, Front. Mar. Sci., 7, 607216, https://doi.org/10.3389/fmars.2020.607216, 2021.